# Shape-directed rotation of homogeneous micromotors via catalytic self-electrophoresis

Ada M. Brooks[1], Mykola Tasinkevych[2], Syeda Sabrina[1], Darrell Velegol[1], Ayusman Sen[3] & Kyle J.M. Bishop [4]

The pursuit of chemically-powered colloidal machines requires individual components that perform different motions within a common environment. Such motions can be tailored by controlling the shape and/or composition of catalytic microparticles; however, the ability to design particle motions remains limited by incomplete understanding of the relevant propulsion mechanism(s). Here, we demonstrate that platinum microparticles move spontaneously in solutions of hydrogen peroxide and that their motions can be rationally designed by controlling particle shape. Nanofabricated particles with $n$-fold rotational symmetry rotate steadily with speed and direction specified by the type and extent of shape asymmetry. The observed relationships between particle shape and motion provide evidence for a self-electrophoretic propulsion mechanism, whereby anodic oxidation and cathodic reduction occur at different rates at different locations on the particle surface. We develop a mathematical model that explains how particle shape impacts the relevant electrocatalytic reactions and the resulting electrokinetic flows that drive particle motion.

[1] Department of Chemical Engineering, Pennsylvania State University, University Park, PA 16802, USA. [2] Centro de Fisica Teórica e Computacional, Departamento de Fisica, Faculdade de Ciências, Universidade de Lisboa, Campo Grande P-1749-016, Lisboa, Portugal. [3] Department of Chemistry, Pennsylvania State University, University Park, PA 16802, USA. [4] Department of Chemical Engineering, Columbia University, New York, NY 10027, USA. Correspondence and requests for materials should be addressed to A.S. (email: asen@psu.edu) or to K.J.M.B. (email: kyle.bishop@columbia.edu)

Catalytic nanomotors[1] harness chemical energy from their environment to power fluid flows and particle motions with emerging opportunities in microrobotics[2,3] and the study of active matter[4,5]. In these contexts, it is often desirable to direct independently the motions of individual motors within heterogeneous ensembles—for example, to guide self-organization in mixtures of counter-rotating spinners[6,7]. Such control cannot be achieved using global fields, which act on all motors simultaneously, but instead requires guidance at the level of the individual particles. For a broad class of self-phoretic motors[8,9], different types of motion can be achieved by controlling the particle symmetry through variations in surface composition and/or geometric shape. For example, axially symmetric motors with broken fore-aft symmetry, such as bimetallic nanorods[1] or platinum Janus spheres[10], enable linear self-propulsion; other motions such as circular[11] and helical trajectories are possible by breaking additional symmetries. While particle symmetry dictates the types of motion allowed (e.g., particle rotation), it says nothing about the quantitative details of such motions (e.g., the speed and direction). The ability to rationally design or program the desired dynamics requires control over particle shape and/or composition as well as knowledge of how such asymmetries direct particle motions. These capabilities could enable the realization of active colloidal ecosystems in which mixtures of different particle species exhibit distinct dynamical behaviors within a common environment to perform collective functions[12–14].

Particle shape provides a particularly attractive medium in which to encode the desired motions of active colloidal particles. In contrast to strategies based on surface patchiness, shape-directed motions can be realized in particles of homogeneous composition, thereby facilitating particle synthesis or fabrication[15]. Moreover, particle shape can be used to control both the type and the extent of asymmetry, thereby enabling the continuous tuning of dynamical behaviors[16]. Perhaps most importantly, the use of shape to direct motion is almost universally applicable to the wide variety of propulsion mechanisms used to power active colloids. Recent examples of shape-directed motions make use of acoustic[17,18], thermocapillary[19], electrokinetic[16,20,21], and vibrational[22,23] energy inputs to propel homogeneous material particles. Similar asymmetries underlie the propulsion of catalytic microrockets[24–27], which direct the release of chemically generated gas bubbles to perform linear and helical motions. By contrast, motors based on the platinum-catalyzed decomposition of hydrogen peroxide—arguably the most studied system—are propelled by self-phoretic flows due to reaction-induced gradients at the particle surface[28,29]. Previous efforts to direct their motion have relied on compositional asymmetries patterned onto particles of isotropic[10] or anisotropic[30–32] shapes. However, it remains unclear if and how shape alone might induce and direct the motions of homogeneous platinum particles through peroxide solutions.

Here, we demonstrate that catalytic micromotors made of solid platinum rotate autonomously in solutions of hydrogen peroxide as directed by asymmetries in their parametric shapes (Fig. 1). Particles are fabricated using projection lithography to form thin plates shaped like twisted stars with $n$-fold rotational symmetry ($C_{nh}$ point group). As anticipated by symmetry considerations, the particles orient parallel to the substrate and rotate steadily about their symmetry axis. Importantly, the direction and speed of rotation is prescribed quantitatively by the type and extent of shape asymmetry; particles can be made to spin clockwise (CW) or counterclockwise (CCW) at a specified rate. Such rotary motions provide a basis for the experimental study of active matter comprised of self-rotating units[6,7,33,34] and may be useful as components of colloidal machines[13,14,35].

To explain the relationship between particle shape and motion, we describe additional experiments and simulations that reveal the likely mechanism underlying catalytic propulsion. Consistent with previous studies of platinum Janus spheres[28,29], we observe that rotation speed decreases monotonically with increasing salt concentration and reverses direction upon addition of a cationic surfactant. These observations support a self-electrophoretic propulsion mechanism analogous to that of bimetallic nanorods[36–38], whereby the anodic oxidation and cathodic reduction of $H_2O_2$ occur at different rates at different locations on the Pt surface[28,29]. The emergence of anodic and cathodic regions on an otherwise homogeneous platinum surface is explained by a

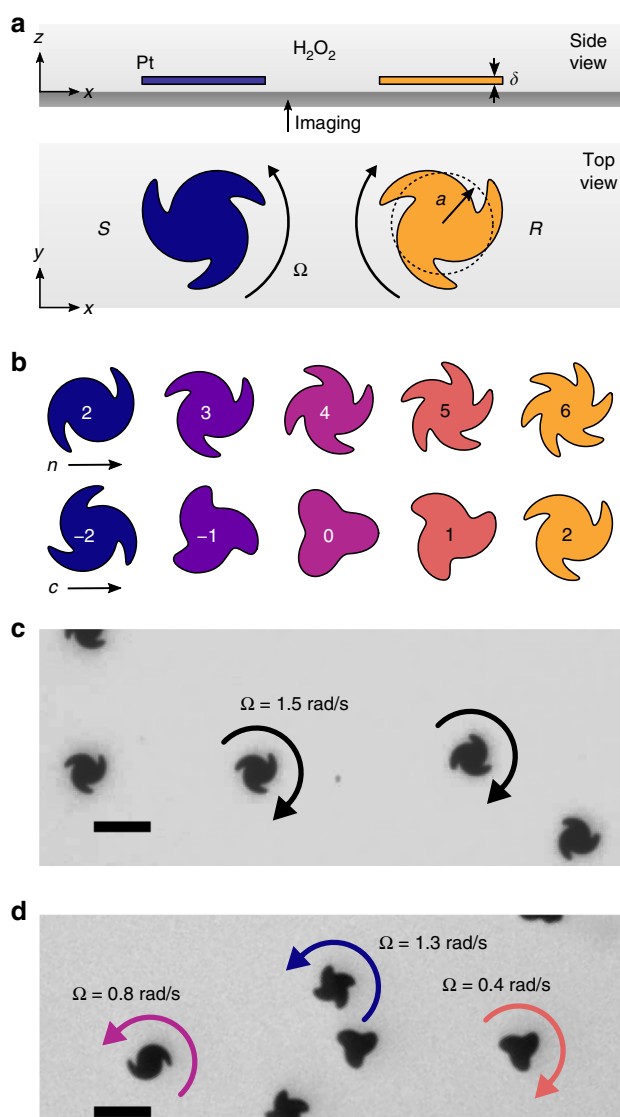

**Fig. 1** Shape-directed particle rotation. **a** Schematic illustration of two platinum disks ($a = 5.77\,\mu m$, $\delta = 100\,nm$) of opposite handedness rotating above a glass substrate in a solution of hydrogen peroxide. **b** The parametric particle shape depends on the number of fins $n$ and the fin asymmetry $c$. **c** Optical microscope image of particles with $n = 3$ and $c = 2$ rotating clockwise at a speed of $\Omega = 1.5 \pm 0.2$ rad s$^{-1}$ in a 10 wt% solution of hydrogen peroxide (see Supplementary Movie 1). **d** Optical microscope image of multiple types of particles rotating in different directions and at different rates in a common environment of 10 wt% hydrogen peroxide (see Supplementary Movie 2). Scale bars are 20 μm

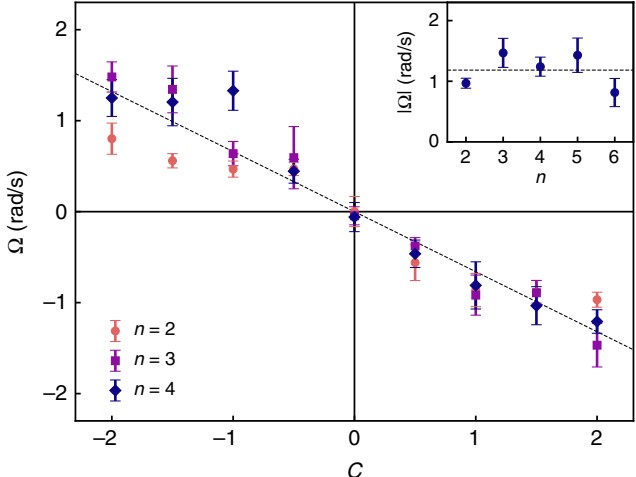

**Fig. 2** Effect of shape asymmetry. The angular velocity $\Omega$ decreases with increasing shape asymmetry $c$ for particles with different numbers of fins $n$. The inset shows the dependence on $n$ for a constant asymmetry of $c = 2$. All experiments were performed in 10 wt% hydrogen peroxide. Error bars denote 95% confidence intervals for the mean velocity based on the analysis of at least 10 particles (see Methods). Dashed lines are only to guide the eye

kinetic mechanism for the electrocatalytic decomposition of hydrogen peroxide[39] on platinum in the limit of transport-controlled reaction rates. With no free parameters, we show that this mechanism is consistent with our experimental observations regarding the direction and speed of rotation as a function of particle shape, peroxide concentration, ionic strength, and particle zeta potential. Looking forward, we discuss how insights from the model can be used to improve the performance of catalytic nanomotors and design shape-directed particle motions and fluid flows with increased functionality.

## Results

**Shape-directed particle motions**. We fabricated plate-like Pt particles ($\delta = 100$ nm thick) with twisted-star shapes using projection photolithography followed by electron beam evaporation (Fig. 1; see Methods). The shape of the particle perimeter was prescribed by the following parametric equations for a plane curve in polar coordinates,

$$r(s) = a[1 + b\cos(ns)] \text{ and } \theta(s) = s + \frac{c}{n}\cos(ns) \quad (1)$$

for $0 < s < 2\pi$ where $r$ denotes the radius of the particle at angle $\theta$, and $s$ is the parameter. These equations define a twisted star with $n$ asymmetric arms (or 'fins') and an average radius of $a = 5.77$ μm (Fig. 1b). The dimensionless parameter $b$ controls the length of each fin, while $c$ determines the degree of fin asymmetry. In our experiments, the fin asymmetry was varied from $c = 0$ to $\pm 2$; the fin length was held constant at $b = 0.3$.

The platinum particles were dispersed in aqueous solutions of hydrogen peroxide and sedimented onto glass slides for imaging by an optical microscope. The particles settled at the surface in a preferred orientation (Fig. 1c) due to a slight warping of the plates introduced during the fabrication process (Supplementary Fig. 1). These observations suggest that the particles were actually chiral ($C_n$ point group) with two distinct stereoisomers denoted $R$ and $S$ by analogy to the chemical convention. In their preferred orientations, $R$ particles correspond to $c > 0$ when viewed from above; $S$ particles to $c < 0$. At the surface, particles rotated steadily at angular speeds of up to 1.5 rad s$^{-1}$ in 10 wt% hydrogen peroxide (Fig. 1c; see also Supplementary Movie 1 and

Supplementary Fig. 2). Within heterogeneous mixtures, particles of different shapes rotated independently in different directions and at different rates within a common environment (Fig. 1d; see also Supplementary Movie 2).

The speed and direction of rotation were determined by particle shape as parameterized by the fin asymmetry $c$ and the number of fins $n$ (Fig. 2). As viewed from above, $R$-particles rotated clockwise ($\Omega < 0$) while $S$-particles rotated counter-clockwise ($\Omega > 0$); achiral particles ($c = 0$) showed no systematic rotation. The angular speed increased monotonically with increasing fin asymmetry as observed previously for acoustically-powered spinners of similar shapes[18]. Interestingly, catalytic spinners rotated in the opposite direction from that of acoustic spinners, highlighting the important role of the propulsion mechanism on shaped-directed particle motions. By contrast to the fin asymmetry, the number of fins $n$ did not significantly influence the speed or direction of particle rotation (Fig. 2, inset).

**Propulsion mechanism**. The propulsion of motors based on the platinum-catalyzed decomposition of hydrogen peroxide has been the subject of considerable debate[9,28], which provides useful context for interpreting the present experiments. The motion of platinum Janus spheres was originally attributed to self-diffusiophoresis[10], whereby asymmetric consumption and production of neutral solutes (namely, H$_2$O$_2$ and dissolved O$_2$) create local concentration gradients that drive interfacial flows due to interactions between the solute(s) and the particle surface[40]. Modeling studies have since predicted that self-diffusiophoresis could propel the motion of catalytic particles with homogeneous surface chemistry and asymmetric shape[41,42]. Alternatively, the motion of Janus spheres was attributed to the asymmetric formation and release of oxygen bubbles[43], which are known to power the motion of larger tubular motors known as microrockets[27,44]. These mechanisms were called into doubt by experiments on platinum Janus motors in electrolyte solutions, which revealed a striking decrease in the propulsion velocity with increasing salt concentration[28,29]. This and other observations supported a different mechanism based on self-electrophoresis[28,29,36] due to spatial variations in the anodic oxidation and cathodic reduction of hydrogen peroxide at the platinum surface,

$$H_2O_2 \rightarrow O_2 + 2H^+ + 2e^- \quad \text{(anode)} \quad (2)$$

$$H_2O_2 + 2H^+ + 2e^- \rightarrow 2H_2O \quad \text{(cathode)} \quad (3)$$

In this scenario, the electric current through the electrolyte from anodic to cathodic surface regions is accompanied by an electric field that drives electrokinetic flows and propels particle motions. The addition of salt increases the conductivity of the electrolyte, thereby reducing the strength of the chemically generated electric field and the speed of particle motions. While the origins of these currents are uncertain, one plausible explanation attributes variations in catalytic activity to gradients in the nanoscale thickness of the platinum layer coating one half of the Janus spheres[29].

To investigate the mechanism of our homogeneous platinum spinners, we first measured how the particle velocity varied with the concentration of hydrogen peroxide fuel (Fig. 3a) and the ionic strength of the surrounding fluid (Fig. 3b). The angular velocity increased monotonically with increasing peroxide concentration in a manner similar to that observed for Pt-coated Janus swimmers (Fig. 3a)[10]. Moreover, the characteristic linear velocity $a\Omega \sim 10$ μm s$^{-1}$ was similar in magnitude to that of other catalytic motors powered by hydrogen peroxide[1,10]. The addition of monovalent salts such as NaCl, KBr, and KNO$_3$ caused a

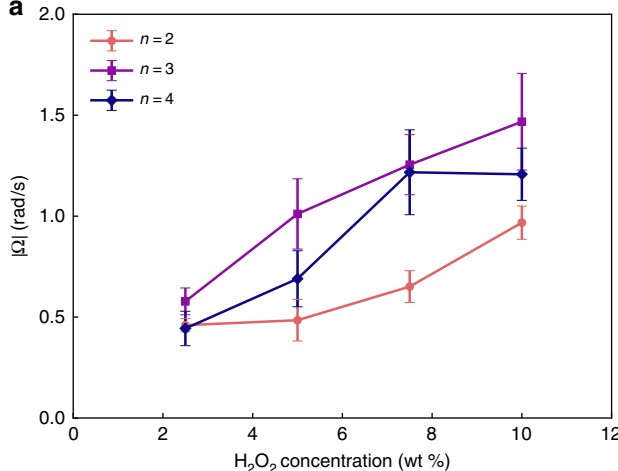

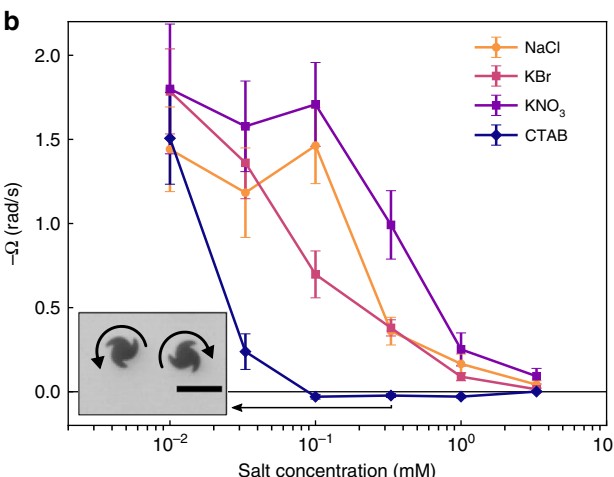

**Fig. 3** Experiments supporting a self-electrophoretic propulsion mechanism. **a** Angular velocity as a function of hydrogen peroxide concentration for particles with shape asymmetry $c = 2$ and different numbers of fins $n$. Error bars represent 95% confidence intervals for the mean velocity. **b** Angular velocity as a function of salt concentration for particles with $n = 3$ fins and shape asymmetry $c = 2$ rotating in 10 wt% $H_2O_2$. The inset shows the rotation reversal with addition of 0.33 mM of the cationic surfactant CTAB (see Supplementary Movie 3); scale bar is 20 μm

reduction in the rotation velocity with increasing concentration (Fig. 3b). Beyond a few millimolar of added salt, particle rotation was no longer observed. As previously argued[28,29], the influence of added salt is not consistent with mechanisms based on neutral self-diffusiophoresis[10] or bubble propulsion[43], which should be largely insensitive to the salt concentration.

Further support for an electrokinetic propulsion mechanism is provided by experiments involving the addition of a cationic surfactant, cetyl triethylammonium bromide (CTAB; Fig. 3b). At low concentrations (less that ca. 0.1 mM), CTAB behaves like a simple salt and reduces the angular velocity of the particles. At higher concentrations, however, the direction of rotation reverses such that $R$-particles rotate in the counterclockwise direction (Fig. 3b; see also Supplementary Movie 3). Such rotation reversal is attributed to the adsorption of CTAB at the particle surface, which changes the zeta potential from negative to positive thereby reversing the direction of electrokinetic flows.

**Model of shape-directed self-electrophoresis.** The putative mechanism outlined above posits the existence of anodic and cathodic regions on the particle surface; however, it does not

explain why these regions exist and how they are influenced by particle shape. Here, we develop a quantitative mathematical model of shape-directed propulsion that provides a plausible explanation for the experimental observations. Our approach combines the standard electrokinetic model[45,46] with the reported kinetic mechanism for the electrocatalytic decomposition of $H_2O_2$ over platinum[39]. The full formulation of the model is detailed in the Methods and the Supplementary Information; here, we focus on the underlying physics, the relevant assumptions, and the key results.

The catalytic decomposition of hydrogen peroxide over platinum occurs by several different kinetic mechanisms, which proceed simultaneously at different rates depending on the reaction conditions[47]. For the present conditions (room temperature, neutral pH, 1 M $H_2O_2$), the dominant mechanism for decomposition is not electrochemical in nature and leads to gradients in neutral species (namely, $H_2O_2$ and $O_2$)[47]. However, these gradients do not contribute significantly to particle propulsion as evidenced by the absence of particle motions at high salt concentrations (>1 mM). Instead, we argue that the dominant mechanism for propulsion relies on the electrocatalytic decomposition of hydrogen peroxide, which contributes only negligibly to the overall rate of peroxide consumption. A similar picture was suggested for the self-electrophoresis of gold-platinum nanorods in hydrogen peroxide, for which the rate of electrocatalytic decomposition was estimated to be only 0.1% of the overall rate[48].

The electrochemical kinetics of anodic oxidation (2) and cathodic reduction (3) of $H_2O_2$ over platinum have been investigated previously to determine the apparent rate laws and possible reaction mechanisms (Fig. 4a)[39]. At high peroxide concentrations ($C_{H_2O_2} > 1$ mM), these reactions proceed with the same rate at the so-called mixed potential, which lies between the reversible (equilibrium) potentials for reactions (2) and (3)[49]. Consequently, both the anodic oxidation and the cathodic reduction of hydrogen peroxide proceed with large overpotentials as approximated by the Tafel equation of electrochemical kinetics[39,50]. This scenario is supported by the measured current densities for peroxide oxidation and reduction as a function of $H_2O_2$ concentration, potential, and pH[51]. Based on this data, Vetter[39] proposed the following expressions for the partial current densities, $i_a$ and $i_c$, at the platinum surface for the anodic and cathodic reactions

$$i_a = k_a C_{HO_2^-} \exp\left(\frac{\alpha_a e}{k_B T}\left(\Phi_p - \Phi_s\right)\right), \quad (4)$$

$$i_c = -k_c C_{H_2O_2} \exp\left(-\frac{(1-\alpha_c)e}{k_B T}\left(\Phi_p - \Phi_s\right)\right). \quad (5)$$

Here, $k_a$ and $k_c$ are the respective rate constants, $C_i$ denotes the concentration of species $i$ at the particle surface, $\alpha_a$ and $\alpha_c$ are charge transfer coefficients, $e$ is the elementary charge, $k_B$ is the Boltzmann constant, and $T$ is the temperature. The Stern layer voltage, $\Phi_p - \Phi_s$, corresponds to the difference between the particle potential $\Phi_p$ and the surface potential $\Phi_s$ at the outer Helmholtz plane[50]. At steady-state, the potential of the particle $\Phi_p$—the mixed potential—is determined by the condition that total current to/from the particle be zero,

$$\int_S (i_a + i_c)dS = 0, \quad (6)$$

where $S$ denotes the particle surface.

The rate law for anodic oxidation (4) suggests that reaction (2) can be divided into two steps: the dissociation of $H_2O_2$ in solution to form $HO_2^-$ and its subsequent consumption at the platinum

surface

$$H_2O_2 \underset{k_{-p}}{\overset{k_p}{\rightleftharpoons}} H^+ + HO_2^- \qquad (7)$$

$$HO_2^- \xrightarrow{k_a} O_2 + H^+ + 2e^- \qquad (8)$$

Vetter suggests a series of elementary steps by which reaction (8) may occur; however, these details are not relevant here[39]. Similarly, the rate law for the cathodic reaction (5) is consistent with a two step reduction of peroxide at the platinum surface followed by the association of $H^+$ and $OH^-$ in solution

$$H_2O_2 + e^- \xrightarrow{k_c} OH + OH^- \qquad (9)$$

$$OH + e^- \rightarrow OH^- \qquad (10)$$

$$2\left(H^+ + OH^- \underset{k_w}{\overset{k_{-w}}{\rightleftharpoons}} H_2O\right) \qquad (11)$$

where reaction (9) is the rate-limiting step[39]. These mechanisms are not firmly established; however, the rate laws (4) and (5) are largely consistent with experimental measurements under conditions relevant to the present experiments—specifically, 0.1 M $H_2O_2$ at neutral to acidic pH values[39,51].

Using the rate laws (4) and (5), we modeled the steady-state concentration profiles for the relevant species ($H^+$, $OH^-$, $HO_2^-$) as a function of distance from a planar platinum surface at the mixed potential (Fig. 4b; see also Methods, Supplementary Note 1, Supplementary Table 1). Importantly, the computed current densities agree with previous experimental measurements[39,51] in the limit as $k_a \rightarrow \infty$, such that the reaction rate is limited by transport of $HO_2^-$ from solution where it is generated by $H_2O_2$ dissociation (Supplementary Note 2). Under these conditions, the concentration of the rate-limiting species $HO_2^-$ increases from zero at the Pt surface to its bulk value in solution over a characteristic length scale $\lambda$ (Fig. 4b)[52]. Analysis of the model reveals that the bulk $HO_2^-$ concentration increases with increasing $H_2O_2$ concentration as $C_\pm^\infty = (K_p C_{H_2O_2}^0)^{1/2}$, where $K_p = k_p/k_{-p}$ is the equilibrium constant for peroxide dissociation (see Methods and Supplementary Note 3). The length scale $\lambda$ is determined by a competition between the generation of $HO_2^-$ in solution and its transport to the surface such that $\lambda = (D_\pm/k_{-p}C_\pm^\infty)^{1/2}$, where $D_\pm = \frac{1}{2}(D_{H^+} + D_{HO_2^-})$ is the average diffusivity of the dominant ions. In experiment, the reaction-diffusion length is estimated to be $\lambda = 139$ nm, which is comparable to the particle thickness ($\delta = 100$ nm) and to the Debye screening length ($\lambda_D = 185$ nm; Table S1).

A key feature of the model is that it predicts the emergence of cathodic and anodic regions on the surface of an otherwise homogeneous platinum particle as a function of particle shape. For this to happen, the anodic and cathodic currents must respond differently to the geometric features of the particle. In the model, the anodic current (4) is transport-limited while the cathodic current (5) is reaction-limited; the former responds to changes in particle shape while the latter does not. To show this, we modeled the steady-state concentration profiles and ionic currents surrounding a thin platinum disk (radius, $a = 41\lambda$; thickness, $\delta = 0.72\lambda$) immersed in a solution of 10 wt% hydrogen peroxide (Fig. 4c, left; see also Methods, Supplementary Figs. 3–5). The high curvature along the particle edge enhances transport of $HO_2^-$ to those locations relative to the particle face, resulting in spatial variations of the anodic current density. By contrast, the

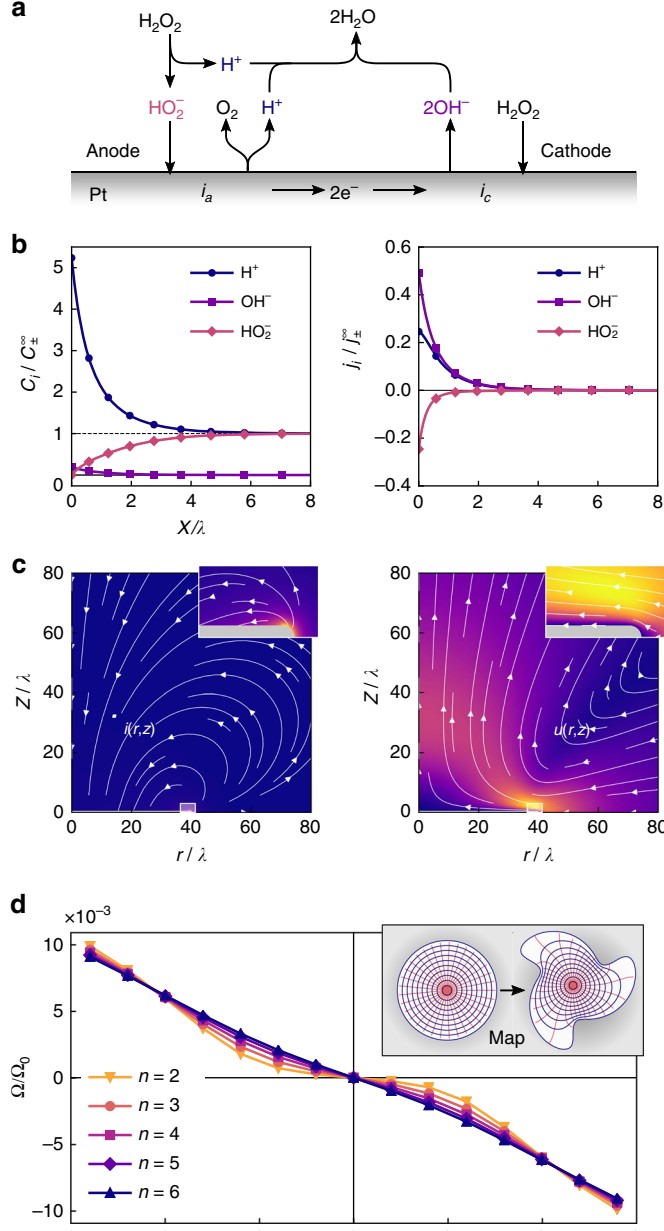

**Fig. 4** Results of the self-electrophoretic propulsion model. **a** Kinetic mechanism for the electrocatalytic decomposition of $H_2O_2$ over platinum. **b** Computed ion concentrations $C_i$ (left) and fluxes $j_i$ (right) as a function of distance $x$ from a planar Pt surface. Concentration is scaled by the bulk concentration $C_\pm^\infty = (K_p C_{H_2O_2}^0)^{1/2}$, distance by the reaction-diffusion length $\lambda = (D_\pm/k_{-p}C_\pm^\infty)^{1/2}$, and flux by the $j_\pm^\infty = D_\pm C_\pm^\infty/\lambda$. **c** Computed current density **i** (left) and fluid velocity **u** (right) around a circular Pt disk of radius $a = 41\lambda$ and thickness $\delta = 0.72\lambda$ centered at the origin. The insets show magnified views of the current and flow near the edge of the disk. The maximum flow velocity is $u_{max} = 0.0082 C_\pm^\infty k_B T\lambda/\eta$, which corresponds to 8.6 μm s$^{-1}$ for the experimental conditions. **d** Computed angular velocity $\Omega$ as a function of the asymmetry parameter $c$ for spinners with $n = 2$–6 fins and $b = 0.3$. Velocities are scaled by $\Omega_0 = C_\pm^\infty k_B Th/\eta a$, which is estimated to be 240 s$^{-1}$ for the experimental conditions. The inset shows the conformal mapping from disk to twisted star used in the calculation

cathodic current density is approximately constant as reaction-induced variations in the peroxide concentration are negligible compared to the bulk concentration. Differences between the anodic and cathodic rates give rise to steady currents through the surrounding electrolyte from the high curvature edges to the low curvature faces. Similar currents are expected for any platinum particle with geometric features comparable to or smaller than the reaction-diffusion length $\lambda$.

Having established a plausible mechanism for the formation of anodic and cathodic regions on the particle surface, the resulting fluid flows and particle motions are anticipated by previous studies of self-electrophoresis[9]. We computed the steady-state fluid flows around a thin platinum disk due to electric forces induced by the electrocatalytic reactions described above (see Methods). The electric field associated with the ionic currents acts on ions within the double layer to drive electrokinetic flows around the particle (Fig. 4c, right). For particles with negative surface potentials, these flows are directed radially inward from the particle edges, at which the electric force density and the fluid velocity are largest. Assuming a surface potential of $\Phi_s = -40$ mV (that of Pt-Au nanorods[53]), the peak flow velocity in 10 wt% peroxide is estimated to be ca. 10 $\mu$m s$^{-1}$. The steady streaming flows induced by the catalytic reactions are qualitatively similar to those driven by external acoustic[18] and electric[16] fields, which are known to drive motions of plate-like particles with asymmetric contours.

The rotation of twisted-star particles can be understood by balancing the net torque on the fluid due to electric forces $L_e$ with the viscous torque due to particle rotation $L_\eta$ such that $L_e = L_\eta$. For thin particles rotating above a planar substrate, the viscous torque can be approximated as $L_\eta = \pi \eta a^4 \Omega / 2h$ where $h \ll a$ is the (unknown) surface separation. We will assume that the separation is comparable to the Debye screening length (i.e., $h = \lambda_D$), which determines the range of electrostatic interactions between the particle and the substrate (Supplementary Note 4). The full computation of the electric torque requires solving a three dimensional, nonlinear transport model on lengths spanning <100 nm (particle thickness) to >10 $\mu$m (particle diameter) to determined the ion currents and the electric field. In light of these difficulties, we adopt a heuristic procedure that uses the axisymmetric solution for the circular disk illustrated in Fig. 4c to roughly approximate the electric torque $L_e$ and thereby the rotation velocity $\Omega$ (see Methods and Supplementary Note 5). Briefly, we map the stress acting at the surface of the disk onto that of a star-shaped particle of equal radius and integrate to obtain the net torque (Fig. 4d, inset). The results of this heuristic procedure agree qualitatively with the experimental measurements (cf. Figs. 2 and 4d). Star-shaped disks are predicted to spin at rates of up to $\Omega = 1$ rad s$^{-1}$ with $R$ spinners rotating CW and $S$ spinners rotating CCW as in experiment. Moreover, the velocity increases with increasing asymmetry $c$ but is largely independent of the number of fins $n$ (Fig. 4d).

The model also helps to explain the dependence of the rotation velocity on the peroxide concentration, the ionic strength, and the particle zeta potential. The predicted rotation speed increases slower than linearly with the concentration of hydrogen peroxide (Supplementary Fig. 6), which is consistent with the experimental results of Fig. 3a. In the model, changes in the peroxide concentration have several competing effects on the chemically driven particle motions. In the absence of added salt, the concentration of the rate-limiting species HO$_2^-$ as well as the pH and ionic strength of the solution are determined by the equilibrium dissociation of H$_2$O$_2$. These quantities are expected to depend sensitively on the presence of additional dissolved species such as carbon dioxide (not considered in the model), which may alter the pH and/or the ionic strength of the solution.

As noted above, the addition of salt increases the electrolyte conductivity, thereby reducing the reaction induced electric fields that drive the electrokinetic flows around the particle (Supplementary Fig. 7). The direction of these flows depends on the sign of the particle surface potential $\Phi_s$, which is assumed equal to the particle zeta potential (Supplementary Fig. 8). For particles with positive zeta potential, predicted fluid flows are directed radially outward from the cathodic face of the particle to its anodic edges, thereby reversing the direction of rotation. Notably, particles with positive surface potentials are predicted to induce faster fluid flow due to enhanced transport of the negatively charged rate limiting species HO$_2^-$ to the particle surface.

Finally, we observe that the transport-limited kinetic model can provide an alternative explanation for the motion of platinum Janus particles in hydrogen peroxide solutions (Supplementary Fig. 9)[10,28,29]. Considering the geometry of the Pt patch, the transport of the rate-limiting reactant HO$_2^-$ is maximal at the edge of the platinum layer along the Janus equator. As a result, this equatorial region becomes more anodic relative to the rest of the Janus cap as previously hypothesized[28,29]. Additionally, the present mechanism can offer an explanation for the reported increase in the particle velocity upon addition of a strong base NaOH at low concentrations[28]. The base acts to decrease the proton concentration, which shifts the dissociation equilibrium of reaction (8) to increase the concentration of the rate-limiting species HO$_2^-$. Further study is required to quantify these and other details of the proposed model. Such studies will require numerical methods that fully describe how the three-dimensional particle shape can direct species transport, electrokinetic flows, and particle motions.

## Discussion

We have shown that solid platinum particles rotate spontaneously in solutions of hydrogen peroxide as directed by their geometric shape alone. Particles with rotational symmetry spin about their axis at a constant rate that depends on the type and extent of fin asymmetry. Perhaps counterintuitively, the angular velocity of the particles was largely insensitive to the number of fins. The non-trivial relationship between particle shape and motion is reproduced and explained by a model of self-electrophoresis that accounts for transport-limitations in the electrocatalytic decomposition of hydrogen peroxide at the platinum surface. Despite uncertainties in the reaction mechanism, the proposed model has no free parameters and provides quantitative predictions that agree with the measured rates[39,51] of peroxide oxidation and reduction at the mixed potential (Supplementary Note 2). This mechanism can also explain the motion of platinum-capped Janus particles (Supplementary Fig. 9).

Particle rotation derives from shape asymmetries on two distinct length scales. First, the sharp edges of the plate-like particles serve to enhance transport of the rate-limiting species HO$_2^-$ and thereby accelerate the anodic half-reaction at those locations. To maintain charge neutrality, ionic charge flows from the anodic edges to the cathodic faces, while the associated electric fields drive fluid flow in the same direction. The electrokinetic flows are guided by a second asymmetry—that of the curved fins—causing the particles to rotate about their axis. We note that these motors are not particularly efficient, since the majority of H$_2$O$_2$ consumed does not contribute to the ionic currents and electric fields that drive fluid flow and particle motion. The predicted rate of electrocatalytic decomposition is ca. 400 times smaller than the overall rate of H$_2$O$_2$ decomposition measured by Brown and Poon[28]. This observation highlights the need for more selective electrocatalysts that could operate more effectively at low fuel concentrations.

The design principles developed here for rotating particles should be helpful in guiding the design of other shape-directed swimmers capable of translational, circular, and helical motions. Rotational motions could be useful for microscale mixing at low Reynolds numbers, while the addition of translation along the rotation axis would aid the design of self-powered microtools[25,54]. By fixing platinum patches of specified geometry to a substrate, one could program microfluidic pumps[55] with asymmetric flow profiles. Perhaps the most exciting (and speculative) opportunities involve motors of different shapes and sizes moving in concert via hydrodynamic, chemical, electrical, and/or steric interactions. Shape-based programming of particle dynamics within such mixtures could provide a basis for multicomponent colloidal machines that assemble to perform dynamic functions.

## Methods

**Fabrication of platinum motors.** Platinum motors were fabricated in the Materials Research Institute of the Pennsylvania State University. Silicon wafers (100 mm, Virginia Semiconductor) were first coated with a 70 nm sacrificial layer of silver using electron beam evaporation in a Semicore E-Gun Thermal Evaporator. A 1 μm layer of LOR5A photoresist was spin-coated onto the wafer, followed by a 0.5 μm layer of Megaposit SPR 3012 photoresist. The motor shapes were designed according to the parametric Eq. (1) using Wolfram Mathematica 11.1 and Tanner L-Edit 16.2. Arrays of such shapes were patterned into the photoresist by projection photolithography using a GCA 8500 Wafer Stepper and developed with Microposit MF CD-26 Developer. After developing, wafers were coated with 100 nm of platinum by electron beam evaporation. The wafers were submerged in acetone for 10 min at room temperature followed by Nano Remover PG at 80 °C for 5 min to strip the photoresist, leaving an array of platinum particles on a layer of silver. The sacrificial silver layer was dissolved in 8 M nitric acid, and the wafer was sonicated to separate the motors. Prior to use, motors were sonicated in 15% hydrogen peroxide (VWR) for 1 h and left in hydrogen peroxide overnight before dispersing in deionized water. Motors were characterized by scanning electron microscopy using the FEI Nova NanoSEM 630 FESEM in Penn State's Materials Characterization Lab (Supplementary Fig. 1).

**Motor experiments.** Motors were mixed with hydrogen peroxide (VWR) diluted to appropriate concentrations and deposited onto glass slides (PTFE slides from Electron Microscopy Sciences). Prior to each experiment, the slide was rinsed in ethanol followed by deionized water. Approximately 90 μL of the particle dispersion was pipetted into a circular well made from a silicone isolation chamber (Electron Microscopy Sciences). Particles in the well were observed using an inverted optical microscope by Carl Zeiss. Videos were captured at 60 frames per second using a Flycam Flea3 digital camera.

Particle motions were tracked manually using Tracker 4.11 (physlets.org). For each condition, at least 10 particles were tracked for 5 s at 6 frames per second. Each particle was tracked at two points fixed to the particle. From the two points, the time evolution of particle orientation was determined. The average angular velocity of each particle was found from the slope of a linear least-squares fit of the particle's orientation versus time (Supplementary Fig. 2). The mean of these slopes along with the 95% confidence intervals were calculated from these data for each experimental condition.

**Governing equations and boundary conditions.** We consider a single platinum particle immersed in a solution of hydrogen peroxide in water with no added salt. The particle moves as a rigid body with linear velocity $\mathbf{U}$ and angular velocity $\mathbf{\Omega}$ through an unbounded quiescent fluid. The distribution of reactive species (namely, $H_2O_2$, $H^+$, $OH^-$, $HO_2^-$) surrounding the particle is governed by conservation equations that account for transport due to convection, diffusion, and migration in the local electric field. At steady-state, the concentration of species $i$ is governed by

$$\nabla \cdot \mathbf{j}_i = \sum_j \nu_{ij} R_j \quad \text{with} \quad \mathbf{j}_i = \mathbf{u}C_i - D_i \nabla C_i + \frac{z_i e D_i}{k_B T} C_i \mathbf{E}, \quad (12)$$

where the species flux $\mathbf{j}_i$ includes contributions due to convection with fluid velocity $\mathbf{u}$, diffusion with diffusivity $D_i$, and migration in the electric field $\mathbf{E} = -\nabla \phi$. The rates of the two dissociation reactions (11) and (7) are given by

$$R_w = k_w C_{H_2O} - k_{-w} C_{H^+} C_{OH^-}, \quad (13)$$

$$R_p = k_p C_{H_2O_2} - k_{-p} C_{H^+} C_{HO_2^-}, \quad (14)$$

which result in the production or consumption of species $i$ in accordance with stoichiometric coefficients $\nu_{ij}$. The electric potential $\phi$ is governed by the Poisson equation

$$-\varepsilon \nabla^2 \phi = \rho_e = e \sum_i z_i C_i, \quad (15)$$

where $\rho_e$ is the charge density, $\varepsilon$ is the permittivity of the fluid, and $z_i$ denotes valence of species $i$. At low Reynolds numbers, the fluid velocity and pressure are well approximated by the Stokes equations,

$$\nabla \cdot \mathbf{u} = 0, \quad (16)$$

$$0 = -\nabla p + \eta \nabla^2 \mathbf{u} + \mathbf{f}_e, \quad (17)$$

which include a volumetric force density, $\mathbf{f}_e = \rho_e \mathbf{E}$, due to the action of the field on the local charge density. In the absence of the dissociation reactions, these equations are often referred to as the standard electrokinetic model[45,46].

Far from the particle, the species concentrations, electric potential, fluid velocity, and pressure approach their bulk values

$$C_i(\mathbf{x}) = C_i^\infty, \quad \phi(\mathbf{x}) = 0, \quad \mathbf{u}(\mathbf{x}) = 0, \quad p(\mathbf{x}) = 0 \quad \text{for } |\mathbf{x}| \to \infty. \quad (18)$$

At the particle surface $\mathcal{S}$, the outward flux of reactive species normal to the interface is balanced by their generation due to the electrochemical reactions detailed above

$$i_a = -2e(\mathbf{n} \cdot \mathbf{j}_{HO_2^-}) = 2e(\mathbf{n} \cdot \mathbf{j}_{H^+}) \quad \text{for } \mathbf{x} \in \mathcal{S}, \quad (19)$$

$$i_c = 2e(\mathbf{n} \cdot \mathbf{j}_{H_2O_2}) = -e(\mathbf{n} \cdot \mathbf{j}_{OH^-}) \quad \text{for } \mathbf{x} \in \mathcal{S}. \quad (20)$$

Here, $\mathbf{n}$ is the unit normal vector directed out from the particle surface. For transport-limited reactions, the flux of $HO_2^-$ is determined implicitly by the additional condition that its concentration approaches zero at the particle surface,

$$C_{HO_2^-}(\mathbf{x}) = 0 \quad \text{for } \mathbf{x} \in \mathcal{S}. \quad (21)$$

The rate of the cathodic reaction is determined by the particle potential $\Phi_p$, which must be computed self-consistently from the zero current condition of Eq. (6). The fluid velocity at the particle surface is governed by the no-slip condition

$$\mathbf{u}(\mathbf{x}) = \mathbf{U} + \mathbf{\Omega} \times \mathbf{x} \quad \text{for } \mathbf{x} \in \mathcal{S}. \quad (22)$$

Finally, the velocities $\mathbf{U}$ and $\mathbf{\Omega}$ are determined by the conditions that there be no net force or torque on the particle surface,

$$\int_{\mathcal{S}} (\boldsymbol{\sigma} + \boldsymbol{\sigma}_e) \cdot \mathbf{n} d\mathcal{S} = 0, \quad (23)$$

$$\int_{\mathcal{S}} \mathbf{x} \times (\boldsymbol{\sigma} + \boldsymbol{\sigma}_e) \cdot \mathbf{n} d\mathcal{S} = 0, \quad (24)$$

where $\boldsymbol{\sigma} = -p\boldsymbol{\delta} + \eta(\nabla \mathbf{u} + \nabla \mathbf{u}^T)$ is the hydrodynamic stress tensor and $\boldsymbol{\sigma}_e = \varepsilon(\mathbf{E}\mathbf{E} - \frac{1}{2}E^2\boldsymbol{\delta})$ is the Maxwell stress tensor, which is related to the electric force as $\mathbf{f}_e = \nabla \cdot \boldsymbol{\sigma}_e$. In Supplementary Note 1, we non-dimensionalize this model in order to identify relevant dimensionless parameters and guide further simplifications; estimates for all model parameters are summarized in Supplementary Table 1.

**Simplifying assumptions.** The analysis of the model above is facilitated by a series of simplifying assumptions summarized here and detailed in the Supplementary Information. First, we focus our attention on large peroxide concentrations, $C_{H_2O_2}^o \gg K_w C_{H_2O}^o / K_p = 4.2$ mM, for which $H^+$ and $HO_2^-$ are the dominant ionic species within the bulk electrolyte. Here, $K_w = k_w/k_{-w} = 1.82 \times 10^{-16}$ M and $K_p = k_p/k_{-p} = 2.40 \times 10^{-12}$ M are the equilibrium dissociation constants for water and peroxide, respectively, at 25 °C. Under these conditions, the equilibrium ion concentrations are

$$C_{H^+}^\infty = C_{HO_2^-}^\infty = \sqrt{K_p C_{H_2O_2}^\infty} + \mathcal{O}(\omega) \text{ and } C_{OH^-}^\infty = 0 + \mathcal{O}(\omega), \quad (25)$$

where $\omega = K_w C_{H_2O}^o / K_p C_{H_2O_2}^o \ll 1$ (Supplementary Note 3). We focus on the limit as $\omega \to 0$, such that the bulk hydroxide concentration is identically zero. In this regime, the characteristic ion concentration increases as the square root of the peroxide concentration.

Second, we assume that the peroxide concentration is everywhere constant and equal to its analytical value, $C_{H_2O_2} = C_{H_2O_2}^o$. This simplification reflects that fact that the overall rate of peroxide consumption at the particle surface is much slower than its rate of delivery by diffusion. This assumption is supported both by previous experiments on platinum Janus particles[28] and by numerical estimates based on the present model (Supplementary Note 2).

Third, we neglect the contribution of fluid convection to the species flux, such that the concentrations $C_i$ and the potential $\phi$ can be solved independently of the fluid velocity $\mathbf{u}$. This assumption is valid when the Péclet number is small—that is,

when $Pe = a^2\Omega/D_{\pm} \ll 1$. For the experimental conditions ($a = 5.77\,\mu m$, $\Omega = 1$ rad s$^{-1}$), the Péclet number is $Pe = 0.005 \ll 1$. The separation of ion transport and fluid motion is useful in adapting results obtained for axially symmetric particles to compute the motions of star-shaped particles used in experiment.

Finally, we assume that the association rate constants $k_{-w}$ and $k_{-P}$ are diffusion-limited. The association rate constant for water is reported to be $k_{-w} = (1.4 \pm 0.2) \times 10^{11}$ M$^{-1}$ s$^{-1}$; the rate constant for peroxide is expected to be similar. To show this, we consider the Debye-Smolucholwski model for diffusion-limited reaction rates between ions[57]

$$k_{-w} = 4\pi d\left(D_+ + D_-\right)\left[\frac{\lambda_B/d}{1 - \exp(-\lambda_B/d)}\right], \quad (26)$$

where $d$ is an ion separation at which the reaction is assumed to occur instantaneously, and $D_{\pm}$ are the diffusion coefficients of the respective ions. The term in brackets describes the acceleration in the collision rate due to Coulombic interactions between the oppositely charged ions where $\lambda_B = e^2/4\pi\varepsilon k_B T$ is the Bjerrum length. For water at 25 °C, the ion diffusivities are $D_+ = 9.31 \times 10^{-9}$ m$^2$/s and $D_- = 5.27 \times 10^{-9}$ m$^2$/s for H$^+$ and OH$^-$ ions, respectively; the Bjerrum length is $\lambda_B = 0.71$ nm, which is considerably larger than the ions themselves. We make the approximation that $d \approx \lambda_B$, which implies that ions approaching closer than $\lambda_B$ are attracted to one another and ultimately react. With the choice that $d = 1.3\lambda_B$, the prediction of Eq. (26) agrees with the measured value quoted above. In the same way, the association rate for peroxide is estimated to be $k_{-P} = 1.0 \times 10^{11}$ M$^{-1}$ assuming that the diffusivity of HO$_2^-$ is equal to that of hydrogen peroxide, $D_{HO_2^-} = 1.43 \times 10^{-9}$ m$^2$/s[58].

**Numerical solution**. With the above assumptions, the model was non-dimensionalized and solved numerically using a commercial finite element solver (COMSOL 5.2a) for the specific case of a circular disk of radius $a$ and thickness $\delta$. Owing to the axial symmetry of the particle, we used a two dimensional, cylindrical coordinate system ($r$, $z$). The ion concentrations, electric potential, and fluid velocity were solved on a large but finite domain: $z > 0$, $r > 0$, and $r^2 + z^2 < 100a^2$. The resulting ion concentrations, electric potential, fluid velocity, and ionic currents are shown graphically in Supplementary Figs. 3–5.

**Conformal mapping**. To approximate the torque $\mathbf{L}_e$ on the twisted star particle, we first compute the stress $\mathbf{e}_z \cdot \boldsymbol{\sigma}$ on the top face of a thin circular disk due to the reaction-induced fluid flows (Fig. 4c). The axial symmetry of the problem implies that the stress is oriented along the radial direction and depends only on the distance $r$ from the center of the particle face—that is, $\mathbf{e}_z \cdot \boldsymbol{\sigma} = \sigma_{zr}(r)\mathbf{e}_r$. For the twisted star particles, we make the heuristic approximation that the stress has an analogous form, $\mathbf{e}_z \cdot \boldsymbol{\sigma} = \sigma_{z\rho}(\rho)\mathbf{e}_\rho$, where $\rho$ is a generalized radial coordinate that ranges from $\rho = 0$ at the particle center to $\rho = 1$ at the particle perimeter. The unit vector $\mathbf{e}_\rho$ in the $\rho$-direction is oriented perpendicular to the particle perimeter and becomes equivalent to $\mathbf{e}_r$ in the limit of weak shape-asymmetry ($b \to 0$ and $c \to 0$). As detailed in Supplementary Note 5, the new coordinate is computed as $\rho = \exp(\nu(\mathbf{r}))$, where $\nu(\mathbf{r})$ satisfies the Poisson equation on a two-dimensional domain bounded by the particle contour $\mathcal{C}$ with a point sink at the origin,

$$\nabla^2\nu = 2\pi\delta(\mathbf{r}) \quad \text{with} \quad \nu(\mathbf{r}) = 0 \text{ for } \mathbf{r} \in \mathcal{C} \quad (27)$$

To estimate the torque, we map the stress on the circular disk onto the twisted star as $\sigma_{z\rho}(\rho) = \sigma_{zr}(r/a)$ and numerically integrate the torque density $\mathbf{r} \times \sigma_{z\rho}\mathbf{e}_\rho$ over the particle face (Fig. 4d). In Supplementary Note 5, we discuss how both the particle shape and the stress distribution $\sigma_{z\rho}(\rho)$ can influence the speed and direction of particle rotation.

## Data availability

The data that support the findings of this study are available from the corresponding authors upon reasonable request.

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

## Acknowledgements

This work was supported in part by the Center for Bio-Inspired Energy Science, an Energy Frontier Research Center funded by the U.S. Department of Energy, Office of Science, Basic Energy Sciences under Award DE-SC0000989. Additional support was provided by the Penn State MRSEC, funded by the National Science Foundation (NSF, DMR-1420620). A.M.B. was supported in part by the National Science Foundation Graduate Research Fellowship Program under Grant DGE1255832. M.T. acknowledges financial support from the Portuguese Foundation for Science and Technology (FCT) under Contract No. IF/00322/2015. We thank Alan West for helpful discussions on electrochemical kinetics.

## Author contributions

A.M.B., S.S., A.S., and K.J.M.B conceived the project and designed the experiments; A.M.B. acquired the data; A.M.B., M.T., S.S., D.V., A.S., and K.J.M.B contributed to the analysis and interpretation of data; A.M.B., M.T., and K.J.M.B. developed the mathematical model and performed the simulations; A.M.B. and K.J.M.B. wrote the paper with input from all authors.

## Additional information

**Competing interests:** The authors declare no competing interests.

