## [Peer Review File · Nature Communications]

Editorial Note: Parts of this peer review file have been redacted as indicated to remove third-party material where no permission to publish could be obtained.

Reviewers' comments:

Reviewer #1 (Remarks to the Author):

This manuscript makes a compelling case for the electrochemical theory of Pt particle propulsion, by applying this theory to elegant experiments on platinum rotors. The manuscript convinces me that the model is very likely to be correct in its essentials, which, as I understand it, are:

- variations in surface curvature produce concentration gradients of chemicals involved in the reaction
- these gradients bias certain regions of the motor towards anodic or cathodic reactions, generating a current and thereby fluid flow
- this fluid flow interacting with the chiral shape of the motor produces rotation.

This paper therefore makes a significant contribution towards our understanding of a wide range of catalytic micromotors, and, as the authors claim, takes us a fair way along the road to the design of more complex, multicomponent micro-machines. It is likely to strongly influence thinking in the field and future experiments and theory on chemically propelled nanomotors. For these reasons I believe that this paper, in a suitably revised form, will be suitable for publication in Nature Communications.

My major concern is that some details of the chemical reaction mechanism are incorrect or less clear than the authors claim. The authors' electrochemical model is based on a 1967 book (ref. 36) that many readers will be unable to access (I was provided with a copy of the relevant section by the publisher). This fails to take into account much subsequent work, where many other mechanisms have been proposed and analyzed. For example, Rui Serra-Mai et al., ACS Applied Materials & Interfaces 2018, 10, 21224-21234 contains a detailed discussion of proposed models and their testable predictions with extensive references. My understanding, from this and related literature, is that there are many potential decomposition mechanisms on Pt, with little consensus, and the likelihood that different mechanisms dominate under different conditions. In addition, from my reading of ref. 36 it is not even clear that Eq. 4-7 and 9-11 should be taken to refer to catalytic H₂O₂ decomposition on platinum. The relevant equations in ref. 36 (4.213 & 4.215, combined in 4.217) refer to electrochemical oxidation and reduction of H₂O₂, presumably with large overpotentials, a physically very different situation from a platinum surface with a mixed potential. The relevant section on catalytic H₂O₂ decomposition (gamma on p641, Eq. 4.216) merely gives the overall electrode reactions, which do not imply a particular mechanism. This point requires a much more detailed and nuanced discussion from the authors, with proper reference to recent literature.

On a related point, I would also like to see a more qualitative discussion of the important parameters and features of the physical model, which is complex. In particular, how sensitive are the results to the precise choice of reaction mechanism? If the physical mechanism is sufficiently generic, this would allay some of my concerns in the previous paragraph, but at the moment this is not clear.

My other comments are relatively minor:

p4: "This characteristic dependence follows from the symmetry of the system and from the fact that the velocity cannot increase without bound upon increasing c ." Why can the velocity not increase without bound (obviously, we're talking about sub-relativistic speeds here)? At the moment, this statement is not an explanation, but simply a restatement of the observation.

In Fig. 2, why not plot the theoretical curves directly on top of the data points? Also, is the evidence for a saturation significant, i.e., what is the p-value for fitting these points to a straight line?

On Fig. 3a, these lines would be fitted reasonably by straight lines through the origin, so I do not believe there is significant evidence for a power-law exponent of 0.63.

In Eq. 8 and elsewhere, the notation with HO_2^- in the subscript is unfortunate, as the minus sign then looks like it is part of the main line. Could this be modified, e.g., by putting HO_2^- as a superscript, or placing the concentration at the end of the line?

The authors' model appears to require a fully conductive Pt surface. Can the authors comment on what would happen with a surface composed of disconnected, individual nanoparticles, as was tested in Ref. 32?

p9: A 'characteristic ion diffusivity'. Various diffusivities have order of magnitude differences, so talking about a characteristic diffusivity is unclear. Ref. 44, in Eq. 38 gives an explicit equation for $\lambda = q^{-1}$ when the two reacting species have different diffusivities.

p10: The authors state that 'The majority of H_2O_2 consumed does not contribute to the ionic current'. This would fully explain their observation, in the SI, that ref. 31 (ref. 6 SI) reports a much higher overall reaction rate than calculations of the current would predict. Hence the authors' additional explanation, that ref. 31 had a very large amount of additional Pt debris would seem to be unnecessary.

p11: Did the "PTFE" slides from electron microscopy sciences have a PTFE coating?

p12: "...these equations are often referred to as the standard electrokinetic model." Referred to by whom? The equations (with and without the dissociation reactions) could do with some appropriate references, possibly to ref. 34 and ref. 44, respectively.

p13: "...the diffusivity of HO_2^- is unknown and assumed equal to that of OH^- for simplicity". This may be wrong by at least an order of magnitude, e.g., ref. 44 cites F. van den Brink, W. Visscher and E. Barendrecht, J. Electroanal. Chem., 1984, 172, 301. to obtain a value of $0.9 \times 10^{-9} \text{ m}^2/\text{s}$ for HO_2^- diffusivity. Whether this value is correct or not, I would expect the diffusivity of HO_2^- to be much lower than the current authors assume, closer to that of O_2 and H_2O_2 than to OH^- .

Eq. 28: Surely the diffusion-limited reaction rate should depend on the size of the ions rather than the Bjerrum length? The authors should cite a peer-reviewed paper where this argument is presented.

Reviewer #2 (Remarks to the Author):

Experimental realizations of self-phoretic catalytic motors so far have been based on asymmetries in the surface coating of the colloids. However, theory suggests that shape asymmetry alone should be enough to drive motion. Here, the authors provide a first experimental realization of such a motor, and provide a theoretical model for the underlying mechanism.

While this is interesting and novel work, I think that in its present form the theoretical part is lacking, and somewhat incongruent with the experimental part. I describe these concerns in the major points below. The authors should address these points in detail in a revised version of the manuscript before I can make a decision.

Major points:

1) The authors dedicate most of the theoretical part to the modelling of the electrochemistry of the system (pages 4-9), and only a couple of paragraphs (pages 9-10) to the modelling of the actual

experimental novelty: the shape asymmetry. The way it is described in the paper, it seems like a wholly new model for the reaction ('transport-limited kinetic model') is introduced, that would give an alternative explanation even for the basic case of Janus colloids (page 10). I think that introducing a new reaction model by applying it to a new experimental phenomenon (shape-directed motion), instead of to a known one, leaves the reader without point of reference, and creates unanswered questions. Can the new reaction model explain well (quantitatively) the motion of Janus colloids? And, conversely, can the alternative previously proposed (not transport-limited) models explain the shape-directed rotation (I would imagine that they can, as rotation should appear generically independently of the precise details of the reaction)? The authors should give clear reasons why the transport-limited model is needed.

- 2) The conformal mapping between a disk and a twisted star is absolutely key to the modelling (after all the asymmetry is the essential ingredient for the rotation), but the authors dedicate literally one or two sentences to it in the main text. Why is this approach valid, and under which conditions? Has it ever been used before in a similar context (there are no references)? How good is the approximation?
- 3) In the end, the authors show that their model can reproduce the observations (Fig 2 vs Fig 4). But I am missing what are the insights achieved from the modelling of the shape-directed motion. What, intuitively, is the mechanism defining the direction of rotation? Why is the number of fins unimportant? Can the authors use the model to give suggestions on how to design shapes resulting in faster-spinning particles?
- 4) In the experimental section, it was said that the velocity increases monotonically with c , and saturates at some point. However, the theoretical result appears to have a maximum at $c=2$ and decrease afterwards, i.e. it is non-monotonic (Fig 4). What is special about $c=2$ that makes the theoretical velocity develop a maximum there? If the rest of parameters are changed, does this maximum shift towards other values of c ? How further would the speed decrease beyond $c=3$? Isn't this a qualitative disagreement between theory and experiment?

Minor points:

- 5) Page 2, paragraph 2. It seems to me that Refs 23-24 should be described separately from 16-22: while 16-22 use external fields and are mostly experimental, 23-24 deal with self-propulsion due purely to shape effects and are theoretical. The existence of such predictions should be explicitly mentioned, because this is precisely what the present paper achieves experimentally. Also the last sentence of the paragraph, "It remains unclear if and how the geometric shape of a homogeneous platinum particle can induce and direct its autonomous motion through hydrogen peroxide solutions." seems to ignore the existence of such predictions.
- 6) Fig 3a: Why a power law fit? Do any of the existing theories for the propulsion of patchy swimmers predict a power law dependence of the propulsion speed on the fuel concentration?
- 7) On page 6, Fig 3c regarding CTAB concentration is referenced, but there is no Fig 3c.
- 8) Typo on the section title on page 6, 'shaped-directed'  'shape-directed'.
- 9) Fig 4c, why are all the field lines broken? Also in Fig S4, bottom left, field lines are missing.

Reviewer #3 (Remarks to the Author):

This manuscript reports about the ability to define the motion behaviour of catalytic micromotors by shape design only. More specifically, the authors convincingly show that the speed and rotation direction can be well set by the degree of asymmetry. The reported work is scientifically sound, systematic and easy to read and understand. The controlled fabrication of the propeller-like motors allows gaining insight into the propulsion mechanism, which can be concluded to be of the self-electrophoretic type. Technically speaking, the work definitely deserves publication but I think the authors need to put their work into a slightly more realistic context.

For instance, the authors say in the introduction "Notably, however, we know of no experiments in which particle shape alone is used to direct the motion of catalytic motors through solutions of chemical fuel."

This is not correct. The motion of a large category of catalytically propelled micromotors, i.e. bubble-propelled microtubular engines, firmly relies on shape-only control. The direction of motion is defined by the axial asymmetry of the microtube, and the trajectory type (e.g. circular, spiral, helical, cork-screw) is directly given by further off-axis shape asymmetries [39]. Such asymmetries were exploited several years ago to create microdrills [ACS Nano 6, 1751 (2012)], something the authors envision in their outlook section. But this does not belong into the outlook, instead it should go into the description of state-of-the art. The speed of bubble-propelled micromotors can be controlled by shape only – even in-operando if soft stimuli-responsive materials are used [Angew. Chem. Int. Ed. 53, 2673 (2014)].

I think these results are important, directly related to the manuscript and should be carefully discussed by the authors in the manuscript. Other parts of the manuscript might need to be amended accordingly.

We gratefully acknowledge the Reviewers for their careful reading of our manuscript and for their constructive recommendations. We have carefully addressed all of the points that were raised, which we believe has strengthened the manuscript. Our point by point response is provided below in blue text; changes to the manuscript are also highlighted in blue.

Reviewer 1

This manuscript makes a compelling case for the electrochemical theory of Pt particle propulsion, by applying this theory to elegant experiments on platinum rotors. The manuscript convinces me that the model is very likely to be correct in its essentials, which, as I understand it, are:

- variations in surface curvature produce concentration gradients of chemicals involved in the reaction
- these gradients bias certain regions of the motor towards anodic or cathodic reactions, generating a current and thereby fluid flow
- this fluid flow interacting with the chiral shape of the motor produces rotation.

This paper therefore makes a significant contribution towards our understanding of a wide range of catalytic micromotors, and, as the authors claim, takes us a fair way along the road to the design of more complex, multicomponent micro-machines. It is likely to strongly influence thinking in the field and future experiments and theory on chemically propelled nanomotors. For these reasons I believe that this paper, in a suitably revised form, will be suitable for publication in *Nature Communications*.

My major concern is that some details of the chemical reaction mechanism are incorrect or less clear than the authors claim. The authors' electrochemical model is based on a 1967 book (ref. 36) that many readers will be unable to access (I was provided with a copy of the relevant section by the publisher). This fails to take into account much subsequent work, where many other mechanisms have been proposed and analyzed. For example, Rui Serra-Mai et al., *ACS Applied Materials & Interfaces*, **10**, 21224-21234 (2018) contains a detailed discussion of proposed models and their testable predictions with extensive references. My understanding, from this and related literature, is that there are many potential decomposition mechanisms on Pt, with little consensus, and the likelihood that different mechanisms dominate under different conditions.

Reply: These concerns are well founded and important. Upon reviewing the cited references on H₂O₂ decomposition, we have arrived at the following working hypothesis. First, the dominant kinetic mechanism for the platinum catalyzed decomposition of hydrogen peroxide is not electrochemical in nature; however, this process does not contribute significantly to motor propulsion. Second, the electrocatalytic mechanism, while negligible to the overall decomposition rate, represents the dominant contribution to motor propulsion. These claims are supported by the following evidence.

The experiments of the Gerischers¹ presented in the Vetter book² (Figure 259) measure the rates of electrocatalytic processes contributing to hydrogen peroxide decomposition. These data are consistent with the transport-limited model we present (using no free parameters; see Supplemental Information). By contrast, the measured rate of the overall peroxide decomposition by Brown & Poon³ is considerably larger (by ca. 400 times) than the electrocatalytic rate predicted by the model. Similar results were observed with bimetallic catalytic nanorods,⁴ where the self-electrophoretic mechanism is well-established. Although negligible to peroxide decomposition, the electrocatalytic processes are sufficient to explain the propulsion of solid Pt spinners as well as Pt Janus spheres (see new Figure S14). The non-electrochemical decomposition processes create gradients of neutral solutes (namely, H₂O₂ and O₂) which could contribute to propulsion via self-diffusiophoresis; however, experiments at higher salt concentrations (> 1 mM) suggest that these contributions are small.

It is indeed unfortunate that the most relevant experimental data (we could find) on the electrocatalytic decomposition of hydrogen peroxide under conditions relevant to motors ($C_{\text{H}_2\text{O}_2} \sim 1$ M) are so difficult to access. As the Reviewer correctly notes, more contemporary work on the mechanisms of peroxide decomposition focus more on the dominant non-electrochemical processes (we now cite Rui Serra-Maia *et al.*). Moreover, electrochemical studies of hydrogen peroxide often focus on its role as a by product or intermediate in oxygen evolution reactions. Electrochemical kinetic studies of hydrogen peroxide at high concentration (> 0.1 M) are hard to find (we now cite another⁵ that builds on the Gerischers' paper¹). We have made extensive revision to the model description and discussion to clarify these points.

In addition, from my reading of ref. 36 it is not even clear that Eq. 4-7 and 9-11 should be taken to refer to catalytic H₂O₂ decomposition on platinum. The relevant equations in ref. 36 (4.213 & 4.215, combined in 4.217) refer to electrochemical oxidation and reduction of H₂O₂, presumably with large overpotentials, a physically very different situation from a platinum surface with a mixed potential. The relevant section on catalytic H₂O₂ decomposition (gamma on p641, Eq. 4.216) merely gives the overall electrode reactions, which do not imply a particular mechanism. This point requires a much more detailed and nuanced discussion from the authors, with proper reference to recent literature.

Reply: The mixed potential lies between the reversible (equilibrium) potentials for peroxide oxidation and reduction. Under these conditions, both oxidation and reduction proceed with large overpotentials as implied by the Tafel equation. This picture is most clearly supported by the Gerischer experiments¹, reproduced below from the Vetter book. The solid lines are predictions of the Tafel equations; these lines intersect at the mixed potential (x -axis) and the exchange current (y -axis), which is approximately $i_+ = i_- = 10^{-4}$ A/cm².

Vetter is unambiguous in stating that the rate of anodic oxidation (4.216a) is given by Eq. (4.211) and that of cathodic reduction (4.216b) is given by Eq. (4.214), which are equivalent to the rate laws used in our model. The development of a mixed potential on platinum in high concentrations of hydrogen peroxide is explained in a similar fashion by Urbach & Bowen.⁵

Redacted

FIG. 259. Partial current density of anodic oxidation (i_+) and cathodic reduction (i_-) of H_2O_2 at bright Pt vs. potential ϵ_h and pH 0, 1 M H_2O_2 with 0,5 M K_2SO_4 supporting electrolyte. [R. and H. GERISCHER: Z. Physik. Chem. (New Series) 6, 178 (1956)].

On a related point, I would also like to see a more qualitative discussion of the important parameters and features of the physical model, which is complex. In particular, how sensitive are the results to the precise choice of reaction mechanism? If the physical mechanism is sufficiently generic, this would allay some of my concerns in the previous paragraph, but at the moment this is not clear.

Reply: A key feature of the model is that it predicts the emergence of cathodic and anodic regions on the surface of an otherwise homogeneous platinum particle as a function of particle shape. For this to happen, the anodic and cathodic currents must respond differently to the geometric features of the particle. In the model, the anodic current is transport-limited while the cathodic current is reaction-limited; the former responds to changes in particle shape while the latter does not. The one feature of the reaction mechanism that is essential to achieving this behavior is the rapid consumption of HO_2^- at the platinum surface implied by the anodic rate law. The details of H_2O_2 reduction are irrelevant; this reaction proceeds at a (nearly) constant rate to balance the anodic current. We have revised the description of the model to clarify its key features and avoid unnecessary details.

My other comments are relatively minor:

p4: “This characteristic dependence follows from the symmetry of the system and from the fact that the velocity cannot increase without bound upon increasing c .” Why can the velocity not increase without bound (obviously, we’re talking about sub-relativistic speeds here)? At the moment, this statement is not an explanation, but simply a restatement of the observation.

Reply: We have revised this sentence.

In Fig. 2, why not plot the theoretical curves directly on top of the data points? Also, is the evidence for a saturation significant, i.e., what is the p-value for fitting these points to a straight line?

Reply: As noted in our response to Reviewer 2, while the formulation of the model is quantitative, our solution of the full 3D problem is highly approximate. Therefore, a direct comparison between the measured and computed rotation rates would be inappropriate. The evidence for saturation is not significant (p -value of 0.36); we now use a simple linear fit.

On Fig. 3a, these lines would be fitted reasonably by straight lines through the origin, so I do not believe there is significant evidence for a power-law exponent of 0.63.

Reply: We agree and have removed the fit.

In Eq. 8 and elsewhere, the notation with HO_2^- in the subscript is unfortunate, as the minus sign then looks like it is part of the main line. Could this be modified, e.g., by putting HO_2^- as a superscript, or placing the concentration at the end of the line?

Reply: We have changed all instances of $C_{\text{HO}_2^-}$ to $C_{\text{HO}_2^-}$ to reduce confusion.

The authors' model appears to require a fully conductive Pt surface. Can the authors comment on what would happen with a surface composed of disconnected, individual nanoparticles, as was tested in Ref. 32?

Reply: If our motors were composed of disconnected nanoparticles, we would not expect self-electrophoresis as the anodic and cathodic currents would balance locally on each nanoparticle.

p9: A 'characteristic ion diffusivity'. Various diffusivities have order of magnitude differences, so talking about a characteristic diffusivity is unclear. Ref. 44, in Eq. 38 gives an explicit equation for $\lambda = q^{-1}$ when the two reacting species have different diffusivities.

Reply: We have clarified that the 'characteristic diffusivity' used to scale the model is the average diffusivity of the dominant ions $D_{\pm} = \frac{1}{2}(D_{\text{H}^+} + D_{\text{HO}_2^-})$.

p10: The authors state that "The majority of H_2O_2 consumed does not contribute to the ionic current". This would fully explain their observation, in the SI, that ref. 31 (ref. 6 SI) reports a much higher overall reaction rate than calculations of the current would predict. Hence the authors' additional explanation, that ref. 31 had a very large amount of additional Pt debris would seem to be unnecessary.

Reply: We agree completely and have clarified the text accordingly.

p11: Did the "PTFE" slides from electron microscopy sciences have a PTFE coating?

Reply: We have confirmed with a representative from Electron Microscopy Sciences that "PTFE" slides have a PTFE coating.

p12: "...these equations are often referred to as the standard electrokinetic model." Referred to by whom? The equations (with and without the dissociation reactions) could do with some appropriate references, possibly to ref. 34 and ref. 44, respectively.

Reply: We have added two references to classic papers that describe the "standard electrokinetic model" in more detail.

p13: "...the diffusivity of HO_2^- is unknown and assumed equal to that of OH^- for simplicity". This may be wrong by at least an order of magnitude, e.g., ref. 44 cites F. van den Brink, W. Visscher and E. Barendrecht, J. Electroanal. Chem., 1984, 172, 301. to obtain a value of $0.9 \times 10^{-9} \text{ m}^2/\text{s}$ for HO_2^- diffusivity. Whether this value is correct or not, I would expect the diffusivity of HO_2^- to be much lower than the current authors assume, closer to that of O_2 and H_2O_2 than to OH^- .

Reply: The cited reference from van den Brink *et al.* provides information on the diffusivity of H_2O_2 , which agrees with the value we quote from Ref. 2 of the Supplementary Information (doi:10.1016/0003-2670(93)80202-V). We agree that the diffusivity of HO_2^- is likely closer to that of H_2O_2 than of OH^- and have updated the model results assuming that $D_{\text{HO}_2^-} = D_{\text{H}_2\text{O}_2}$.

Eq. 28: Surely the diffusion-limited reaction rate should depend on the size of the ions rather than the Bjerrum length? The authors should cite a peer-reviewed paper where this argument is presented.

Reply: We now cite the original paper from Debye on diffusion-limited association rates between ions; we clarify the approximations made in the Methods.

Reviewer 2

Experimental realizations of self-phoretic catalytic motors so far have been based on asymmetries in the surface coating of the colloids. However, theory suggests that shape asymmetry alone should be enough to drive motion. Here, the authors provide a first experimental realization of such a motor, and provide a theoretical model for the underlying mechanism.

While this is interesting and novel work, I think that in its present form the theoretical part is lacking, and somewhat incongruent with the experimental part. I describe these concerns in the major points below. The authors should address these points in detail in a revised version of the manuscript before I can make a decision.

Major points:

1) The authors dedicate most of the theoretical part to the modelling of the electrochemistry of the system (pages 4-9), and only a couple of paragraphs (pages 9-10) to the modelling of the actual experimental novelty: the shape asymmetry. The way it is described in the paper, it seems like a wholly new model for the reaction ('transport-limited kinetic model') is introduced, that would give an alternative explanation even for the basic case of Janus colloids (page 10). I think that introducing a new reaction model by applying it to a new

experimental phenomenon (shape-directed motion), instead of to a known one, leaves the reader without point of reference, and creates unanswered questions. Can the new reaction model explain well (quantitatively) the motion of Janus colloids? And, conversely, can the alternative previously proposed (not transport-limited) models explain the shape-directed rotation (I would imagine that they can, as rotation should appear generically independently of the precise details of the reaction)? The authors should give clear reasons why the transport-limited model is needed.

Reply: We thank the Reviewer for these constructive criticisms. We followed the Reviewer’s recommendation and applied the proposed model to the well-studied system of a platinum-coated Janus sphere (Fig. S14). The results are described within a new section and figure of the Supplementary Information. In short, we find that our model agrees well with experimental results. Moreover, we emphasize that this model has zero tunable parameters.

In our view, there exists no well-established model that explains the experimental observations on platinum motors in hydrogen peroxide. The most common theoretical description is based on self-diffusiophoresis and uses the diffusiophoretic mobility as a fitting parameter. Shape-directed propulsion based on self-diffusiophoresis has been described theoretically (e.g., Michelin & Lauga⁶); however, it is not directly relevant to the present experiments. As argued by others (Brown & Poon,³ Ebbens *et al.*⁷), this mechanism is inconsistent with the experimental observations of Pt-Janus motors in electrolyte solutions. Instead, the experimental evidence supports a self-electrokinetic mechanism for the propulsion of Pt-Janus spheres. Models of self-electrophoretic propulsion are very similar to that described here with one important distinction: the distribution of anodic and cathodic regions on the particle surface is assumed *a priori* and often in an *ad hoc* manner. Consequently, these models offer little guidance in understanding how particle shape can influence the existence and location of anodic/cathodic regions in the first place.

Our discussion therefore focuses on the most novel feature of our proposed model—namely, its prediction of anodic and cathodic surface regions due to asymmetries in particle shape. We have extensively revised our presentation of the model to clarify these points and to better communicate why a “new” model is needed in the first place.

2) The conformal mapping between a disk and a twisted star is absolutely key to the modelling (after all the asymmetry is the essential ingredient for the rotation), but the authors dedicate literally one or two sentences to it in the main text. Why is this approach valid, and under which conditions? Has it ever been used before in a similar context (there are no references)? How good is the approximation?

Reply: The full numerical implementation of the model in three dimensions spanning lengths < 100 nm to > 10 μ m is challenging (for us) and remains the subject of an ongoing research effort. In lieu of these results, we adopt an approximate heuristic to estimate the speed and direction of particle rotation using the more detailed simulation of the circular disk. This conformal mapping approach is now described in more detail in the Methods and in the Supplementary Information. We are not aware of other studies where this approach has been employed in a similar context.

In a new section of the Supplementary Information (Section 2.6), we investigate how the stress distribution and the particle shape contribute to the net torque that drives particle rotation. For twisted star particles, we find that the direction of rotation depends on the radial stress distribution in qualitative agreement with the present experiments and with previous experiments on the acoustic rotation of gold micro-plates of similar shapes.

3) In the end, the authors show that their model can reproduce the observations (Fig 2 vs Fig 4). But I am missing what are the insights achieved from the modelling of the shape-directed motion. What, intuitively, is the mechanism defining the direction of rotation? Why is the number of fins unimportant? Can the authors use the model to give suggestions on how to design shapes resulting in faster-spinning particles?

Reply: As detailed in a new section of the Supplementary Information (Section 2.6), an intuitive explanation for the direction of rotation and its dependence on particle shape remains elusive. Nevertheless, it is possible to use the present model to suggest particle shapes that spin faster than others. One obvious solution is to increase the asymmetry parameter c .

4) In the experimental section, it was said that the velocity increases monotonically with c , and saturates at some point. However, the theoretical result appears to have a maximum at $c=2$ and decrease afterwards, i.e. it is non-monotonic (Fig 4). What is special about $c = 2$ that makes the theoretical velocity develop a maximum there? If the rest of parameters are changed, does this maximum shift towards other values of c ? How further would the speed decrease beyond $c = 3$? Isn't this a qualitative disagreement between theory and experiment?

Reply: As noted by Reviewer 1, the experimental evidence for saturation with increasing c is not statistically significant. Moreover, the predicted maximum at $c = 2$ was caused by an error in our previous analysis. In mapping the stress from the disk to the star, we included the contributions from the disk edge, which was then stretched over regions of the star face. We now exclude these contributions and map the stress on the face of the disk to that on the face of the star. The predicted rotation velocity increases monotonically with increasing c within the range of interest.

Minor points:

5) Page 2, paragraph 2. It seems to me that Refs 23-24 should be described separately from 16-22: while 16-22 use external fields and are mostly experimental, 23-24 deal with self-propulsion due purely to shape effects and are theoretical. The existence of such predictions should be explicitly mentioned, because this is precisely what the present paper achieves experimentally. Also the last sentence of the paragraph, "It remains unclear if and how the geometric shape of a homogeneous platinum particle can induce and direct its autonomous motion through hydrogen peroxide solutions." seems to ignore the existence of such predictions.

Reply: Refs 23-24 predict shape-directed self-diffusiophoresis; we demonstrate shape-directed

self-electrophoresis. We discuss the distinction in new text within the “Propulsion Mechanism” section of the main text.

6) Fig 3a: Why a power law fit? Do any of the existing theories for the propulsion of patchy swimmers predict a power law dependence of the propulsion speed on the fuel concentration?

Reply: Given their uncertainty, these data are likely to be consistent with several possible models. We have removed the fit to avoid biasing the reader’s interpretation of the experimental results.

7) On page 6, Fig 3c regarding CTAB concentration is referenced, but there is no Fig 3c.

Reply: Thank you. We intended to reference Figure 3b. We have corrected this error.

8) Typo on the section title on page 6, ‘shaped-directed’ → ‘shape-directed’.

Reply: Thank you. We have corrected the typo.

9) Fig 4c, why are all the field lines broken? Also in Fig S4, bottom left, field lines are missing.

Reply: The streamplot function in Matplotlib provides an approximately uniform density of streamlines, which necessitates breaking them into segments of finite length. In Figure S4, there are no hydroxide currents outside the reaction diffusion layer; we have clarified this point in the caption.

Reviewer 3

This manuscript reports about the ability to define the motion behaviour of catalytic micro-motors by shape design only. More specifically, the authors convincingly show that the speed and rotation direction can be well set by the degree of asymmetry. The reported work is scientifically sound, systematic and easy to read and understand. The controlled fabrication of the propeller-like motors allows gaining insight into the propulsion mechanism, which can be concluded to be of the self-electrophoretic type. Technically speaking, the work definitely deserves publication but I think the authors need to put their work into a slightly more realistic context.

For instance, the authors say in the introduction “Notably, however, we know of no experiments in which particle shape alone is used to direct the motion of catalytic motors through solutions of chemical fuel.”

This is not correct. The motion of a large category of catalytically propelled micromotors, i.e. bubble-propelled microtubular engines, firmly relies on shape-only control. The direction of motion is defined by the axial asymmetry of the microtube, and the trajectory type (e.g., circular, spiral, helical, cork-screw) is directly given by further off-axis shape asymmetries [39]. Such asymmetries were exploited several years ago to create microdrills [*ACS Nano* **6**, 1751 (2012)], something the authors envision in their outlook section. But this does not belong into the outlook, instead it should go into the description of state-of-the art. The speed of bubble-propelled micromotors can be controlled by shape only – even in-operando if soft stimuli-responsive materials are used [*Angew. Chem. Int. Ed.* **53**, 2673 (2014)].

I think these results are important, directly related to the manuscript and should be carefully discussed by the authors in the manuscript. Other parts of the manuscript might need to be amended accordingly.

Reply: The Reviewer is absolutely correct on this point: bubble-propelled microtubular engines are very much directed by shape. We have revised the introduction accordingly and added citations to these and other references on this class of catalytic motor. Additionally, we have revised the conclusion to acknowledge the tubular microdrills designed in 2012.

References

1. Gerischer, R. & Gerischer, H. Über die katalytische Zersetzung von Wasserstoffsuperoxyd an metallischem Platin. *Zeitschrift für Phys. Chemie* **6**, 178–200 (1956).
2. Vetter, K. J. *Electrochemical Kinetics* pp. 635–642 (Academic Press, New York, 1967).
3. Brown, A. & Poon, W. Ionic effects in self-propelled Pt-coated Janus swimmers. *Soft Matter*, 4016–4027 (2014).
4. Wang, W., Chiang, T. Y., Velegol, D. & Mallouk, T. E. Understanding the efficiency of autonomous nano- and microscale motors. *Journal of the American Chemical Society* **135**, 10557–10565 (2013).

5. Urbach, H. B. & Bowen, R. J. Behaviour of the oxygen-peroxide couple on platinum. *Electrochim. Acta* **14**, 927–940 (1969).
6. Michelin, S. & Lauga, E. Autophoretic locomotion from geometric asymmetry. *Eur. Phys. J. E* **38**, 7 (2015).
7. Ebbens, S., Gregory, D. A., Dunderdale, G., Howse, J. R., Ibrahim, Y., Liverpool, T. B. & Golestanian, R. Electrokinetic effects in catalytic platinum-insulator Janus swimmers. *Europhys. Lett.* **106**, 58003 (2014).

REVIEWERS' COMMENTS:

Reviewer #1 (Remarks to the Author):

I note that, in agreement with the editor, I am submitting a rather more minimal report than I would normally hope to provide. This is due to a lack of time on my part, and is based on a rapid read-through of the revised manuscript and replies, which did not allow me, in particular, to really assess the detailed mathematics relating to the conformal mapping in the SI.

Nevertheless, from this I am satisfied that the authors have throughout made the assumptions and generality of their model much clearer, and have otherwise addressed all of my concerns. My only suggestion is that the authors feature something like this sentence from their reply to me more prominently in the manuscript itself: "For this to happen, the anodic and cathodic currents must respond differently to the geometric features of the particle. In the model, the anodic current is transport-limited while the cathodic current is reaction-limited; the former responds to changes in particle shape while the latter does not." In my opinion, this is the central principle of their mechanism, and so deserves to be hammered home a bit more. Otherwise, I believe that this excellent paper is now in a suitable form for publication in Nature Communications.

Reviewer #2 (Remarks to the Author):

The authors have satisfactorily addressed my comments and I recommend publication of the paper.

Reviewer #3 (Remarks to the Author):

The authors have properly responded to my comments. They should add the author list to Ref. 25. Once this is completed, I recommend publication of the manuscript in Nature Comm.

Reviewer 1

I note that, in agreement with the editor, I am submitting a rather more minimal report than I would normally hope to provide. This is due to a lack of time on my part, and is based on a rapid read-through of the revised manuscript and replies, which did not allow me, in particular, to really assess the detailed mathematics relating to the conformal mapping in the SI.

Nevertheless, from this I am satisfied that the authors have throughout made the assumptions and generality of their model much clearer, and have otherwise addressed all of my concerns. My only suggestion is that the authors feature something like this sentence from their reply to me more prominently in the manuscript itself: “For this to happen, the anodic and cathodic currents must respond differently to the geometric features of the particle. In the model, the anodic current is transport-limited while the cathodic current is reaction-limited; the former responds to changes in particle shape while the latter does not.” In my opinion, this is the central principle of their mechanism, and so deserves to be hammered home a bit more. Otherwise, I believe that this excellent paper is now in a suitable form for publication in Nature Communications.

Reply: Thank you. We have incorporated this point from our response into the main text (page 6).

Reviewer 2

The authors have satisfactorily addressed my comments and I recommend publication of the paper.

Reply: Thank you.

Reviewer 3

The authors have properly responded to my comments. They should add the author list to Ref. 25. Once this is completed, I recommend publication of the manuscript in Nature Comm.

Reply: Thank you. We have corrected Reference 25.